# A type III-A CRISPR-Cas system employs degradosome nucleases to ensure robust immunity

**Lucy Chou-Zheng, Asma Hatoum-Aslan\***

Department of Biological Sciences, The University of Alabama, Tuscaloosa, United States

**Abstract** CRISPR-Cas systems provide sequence-specific immunity against phages and mobile genetic elements using CRISPR-associated nucleases guided by short CRISPR RNAs (crRNAs). Type III systems exhibit a robust immune response that can lead to the extinction of a phage population, a feat coordinated by a multi-subunit effector complex that destroys invading DNA and RNA. Here, we demonstrate that a model type III system in *Staphylococcus epidermidis* relies upon the activities of two degradosome-associated nucleases, PNPase and RNase J2, to mount a successful defense. Genetic, molecular, and biochemical analyses reveal that PNPase promotes crRNA maturation, and both nucleases are required for efficient clearance of phage-derived nucleic acids. Furthermore, functional assays show that RNase J2 is essential for immunity against diverse mobile genetic elements originating from plasmid and phage. Altogether, our observations reveal the evolution of a critical collaboration between two nucleic acid degrading machines which ensures cell survival when faced with phage attack.

DOI: https://doi.org/10.7554/eLife.45393.001

**\*For correspondence:**
ahatoum@ua.edu

**Competing interests:** The authors declare that no competing interests exist.

## Introduction

Nearly all known *archaea* and about half of *bacteria* possess adaptive immune systems composed of clusters of regularly-interspaced short palindromic repeats (CRISPRs) and CRISPR-associated (Cas) proteins (*Makarova et al., 2015*). CRISPR-Cas systems utilize small CRISPR RNAs (crRNAs) to recognize and destroy prokaryotic viruses (phages) (*Barrangou et al., 2007*; *Brouns et al., 2008*) and other mobile genetic elements (*Bikard et al., 2012*; *Marraffini and Sontheimer, 2008*). The CRISPR-Cas immune response occurs in three steps: adaptation, crRNA biogenesis, and interference (reviewed in *Klompe and Sternberg, 2018*). During adaptation, Cas proteins capture short segments of foreign nucleic acids (24–40 nucleotides (nts) in length) and integrate them as 'spacers' into the CRISPR locus in between repeat sequences of similar lengths (*Mojica et al., 2005*; *Yosef et al., 2012*) in order to record a molecular memory of past invaders. During crRNA biogenesis, the repeats and spacers are transcribed into a long precursor crRNA, which is subsequently processed to generate mature crRNAs that each contain a single spacer sequence. Mature crRNAs combine with one or more Cas proteins to form an effector complex, which during interference, senses and degrades 'protospacers', invading nucleic acids complementary to the crRNA. Although all CRISPR-Cas systems adhere to this general pathway, they exhibit striking diversity in their *cas* gene composition and corresponding mechanisms of action. Accordingly, the current classification scheme divides these systems into two classes, six types (I-VI), and dozens of subtypes (*Koonin et al., 2017*; *Makarova et al., 2015*). Class one systems (types I, III, and IV) encode multi-subunit effector complexes, while Class two systems (types II, V, and VI) possess single subunit effectors. Although Class two systems have garnered significant attention over the past decade due to their versatile genetic applications (*Klompe and Sternberg, 2018*), Class one systems are more widespread in nature, and

**eLife digest** Just as humans are susceptible to viruses, bacteria have their own viruses to contend with. These viruses – known as phages – attach to the surface of bacterial cells, inject their genetic material, and use the cells' enzymes to multiply while destroying their hosts.

To defend against a phage attack, bacteria have evolved a variety of immune systems. For example, when a bacterium with an immune system known as CRISPR-Cas encounters a phage, the system creates a 'memory' of the invader by capturing a small snippet of the phage's genetic material. The pieces of phage DNA are copied into small molecules known as CRISPR RNAs, which then combine with one or more Cas proteins to form a group called a Cas complex. This complex patrols the inside of the cell, carrying the CRISPR RNA for comparison, similar to the way a detective uses a fingerprint to identify a criminal. Once a match is found, the Cas proteins chop up the invading genetic material and destroy the phage.

There are several different types of CRISPR-Cas systems. Type III systems are among the most widespread in nature and are unique in that they provide a nearly impenetrable barrier to phages attempting to infect bacterial cells. Medical researchers are exploring the use of phages as alternatives to conventional antibiotics and so it is important to find ways to overcome these immune responses in bacteria. However, it remains unclear precisely how Type III CRISPR-Cas systems are able to mount such an effective defense.

Chou-Zheng and Hatoum-Aslan used genetic and biochemical approaches to study the Type III CRISPR-Cas system in a bacterium called *Staphylococcus epidermidis*. The experiments showed that two enzymes called PNPase and RNase J2 played crucial roles in the defense response triggered by the system. PNPase helped to generate CRISPR RNAs and both enzymes were required to help to destroy genetic material from invading phages.

Previous studies have shown that PNPase and RNase J2 are part of a machine in bacterial cells that usually degrades damaged genetic material. Therefore, these findings show that the Type III CRISPR-Cas system in *S. epidermidis* has evolved to coordinate with another pathway to help the bacteria survive attack from phages. CRISPR-Cas immune systems have formed the basis for a variety of technologies that continue to revolutionize genetics and biomedical research. Therefore, along with aiding the search for alternatives to antibiotics, this work may potentially inspire the development of new genetic technologies in the future.

DOI: https://doi.org/10.7554/eLife.45393.002

of these, type III systems are believed to be the most closely related to the common ancestor from which all other CRISPR types have evolved (*Mohanraju et al., 2016*).

*Staphylococcus epidermidis* RP62a harbors a well-established model type III-A CRISPR-Cas system (*Figure 1A*), here onward referred to as CRISPR-Cas10. This system encodes three spacers (*spc*) and nine CRISPR-associated proteins (Cas and Csm) that can prevent the transfer of a conjugative plasmid (*Marraffini and Sontheimer, 2010*) and stave off phage infection (*Maniv et al., 2016*). This system employs an elaborate transcription-dependent mechanism of defense that exerts exquisite spatial and temporal control over the immune response. Defense begins with crRNA biogenesis (*Figure 1B*), during which the Cas6 endoribonuclease cleaves the precursor crRNA within repeat sequences to generate 71 nt intermediate crRNAs (*Hatoum-Aslan et al., 2011*; *Hatoum-Aslan et al., 2014*). These intermediates are subsequently trimmed on their 3'-ends to generate mature species that range in length from 31 to 43 nts (*Hatoum-Aslan et al., 2013*). Mature crRNAs combine with Cas10, Csm2, Csm3, Csm4, and Csm5 to form the Cas10-Csm effector complex (*Hatoum-Aslan et al., 2013*) (*Figure 1C*). During interference, this complex, in conjunction with the accessory ribonuclease Csm6, wage a two-pronged attack against foreign DNA and RNA. Interference is triggered by the binding of the crRNA to the targeted transcript (*Goldberg et al., 2014*), an event that leads to the activation of at least three nucleases: Cas10 cleaves the non-template (coding) DNA strand (*Samai et al., 2015*), each Csm3 subunit slices RNA within the protospacer region (*Samai et al., 2015*; *Staals et al., 2014*; *Tamulaitis et al., 2014*), and Csm6 degrades nonspecific transcripts in the vicinity (*Foster et al., 2018*; *Jiang et al., 2016*). In related type III-A systems, target RNA binding has also been shown to trigger two additional functions of Cas10: the cleavage of non-

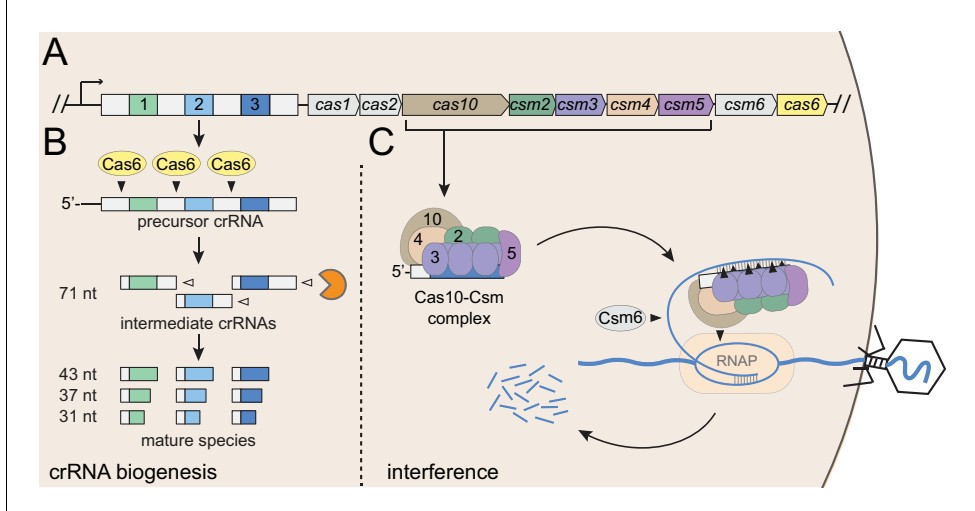

**Figure 1.** Type III-A CRISPR-Cas defense in *S. epidermidis* RP62a. (**A**) The type III-A CRISPR-Cas system in *S. epidermidis* (known as CRISPR-Cas10) encodes three spacers (colored squares), four repeats (white squares) and nine CRISPR-associated (*cas* and *csm*) genes. (**B**) During crRNA biogenesis, the repeat-spacer array is transcribed into a precursor crRNA, which is subsequently processed in two steps. First, the Cas6 endoribonuclease cleaves within repeat sequences to yield intermediate crRNAs that are ~71 nucleotides in length. These intermediates are then subjected to a second maturation step in which their 3'-ends are trimmed to generate mature crRNAs that range from 31 to 43 nucleotides in length. Mature crRNAs combine with Cas10, Csm2, Csm3, Csm4, and Csm5 to form the Cas10-Csm complex. (**C**) Interference is triggered when the Cas10-Csm complex binds foreign transcripts that bear complementarity to the crRNA. During interference, invading DNA and RNA are destroyed–while RNA is shredded by each Csm3 subunit in the complex, the DNA coding strand is cleaved by Cas10. An accessory nuclease, Csm6, also participates in the immune response by degrading invader-derived transcripts. Filled triangles indicate cleavage events catalyzed by CRISPR-associated proteins, and open triangles indicate cleavage events catalyzed by non-Cas nucleases. RNAP, RNA polymerase.

DOI: https://doi.org/10.7554/eLife.45393.003

specific single-stranded DNA by its HD domain (*Kazlauskiene et al., 2016*; *Liu et al., 2017*), and the generation of cyclic oligoadenylates by its Palm polymerase domain (*Kazlauskiene et al., 2017*; *Niewoehner et al., 2017*). The cyclic oligoadenylates act as second messengers which bind and further stimulate Csm6, thus accelerating its activity near the site of foreign nucleic acid detection. The combined activities of the Cas and Csm nucleases, together with the high tolerance for mismatches between the crRNA and protospacer pairing (*Pyenson et al., 2017*), contribute to a particularly powerful immune response that can lead to phage extinction with a single targeting spacer (*Bari et al., 2017*; *Pyenson et al., 2017*).

The robust protection conferred by type III CRISPR-Cas systems motivated us to pose the question: Do these systems conspire with other cellular pathways to ensure a successful defense? Indeed, our previous work showed that the Cas10-Csm complex, when expressed in its native *S. epidermidis* host, co-purifies with several cellular nucleases in trace amounts (*Walker et al., 2017*). Furthermore, in a reconstituted system, Csm5 can bind and stimulate one of these nucleases, PNPase (polynucleotide phosphorylase), a highly-conserved enzyme that is responsible for the processing and degradation of cellular RNAs in bacteria and eukaryotes (*Cameron et al., 2018*). This observation lead us to propose a model wherein PNPase and other non-Cas nucleases are likely responsible for crRNA maturation (*Walker et al., 2017*). However, the extent to which the success of CRISPR-Cas10 defense relies upon the activities of these non-Cas nucleases remains unknown.

In bacteria, PNPase can act independently, or associate with other enzymes (including nucleases and helicases) to form an RNA degrading machine called the degradosome (*Cameron et al., 2018*). In this study, we posited the hypothesis that PNPase and other degradosome-associated nucleases are essential for CRISPR-Cas10 immunity. To test this, we deleted the genes encoding PNPase and Ribonuclease (RNase) J2, a second degradosome nuclease that co-purifies with the Cas10-Csm

complex (*Walker et al., 2017*). The single and double mutants were then assayed for defects in crRNA biogenesis and interference. We discovered that PNPase is indeed required for efficient crRNA maturation, but RNase J2 is not. In contrast, while PNPase appears to be dispensable for CRISPR-Cas10 function, RNase J2 is essential to block the transfer of a conjugative plasmid and defend against two unrelated phages, therefore establishing a general requirement for RNase J2 in CRISPR-Cas10 immunity. We also used quantitative PCR to track the accumulation of nucleic acids during an active phage infection, and found that both PNPase and RNase J2 are required for the efficient clearance of phage-derived DNA and transcripts. It has been well documented by others that RNase J2 promotes cellular RNA degradation by binding and stimulating the ribonuclease activity of its paralog RNase J1 (*Hausmann et al., 2017*; *Linder et al., 2014*; *Mathy et al., 2007*; *Mathy et al., 2010*; *Raj et al., 2018*). Here, we show that *S. epidermidis* RNase J2 also promotes robust DNA degradation through this same mechanism. Altogether, our results support a model for CRISPR-Cas10 immunity in which two steps of the pathway, crRNA biogenesis and interference, rely upon the activities of degradosome-associated nucleases, thus revealing the evolution of a critical collaboration that ensures cell survival when faced with phage infection.

## Results

### PNPase promotes crRNA maturation, but RNase J2 does not

PNPase is a processive exonuclease that degrades RNA and DNA in the 3′−5′ direction, and also catalyzes the reverse reaction in which nucleotide diphosphates are added back onto the 3′-end of the substrate (*Cameron et al., 2018*; *Cardenas et al., 2009*; *Walker et al., 2017*). Our previous work showed that in a purified system, Csm5 can bind to PNPase and stimulate its nucleolytic activity while repressing its polymerization function (*Walker et al., 2017*). Since the deletion of Csm5 leads to loss of crRNA maturation in vivo (*Hatoum-Aslan et al., 2011*; *Hatoum-Aslan et al., 2014*), and since we were unable to locate a nucleolytic active site in Csm5 after extensive mutagenic and biochemical analyses, the observed physical and functional interactions between Csm5 and PNPase lead us to propose a model in which Csm5 recruits and stimulates the catalytic activity of PNPase to carry out crRNA maturation (*Walker et al., 2017*). Here, to lend further support to this model, we created in-frame deletions of *pnp* in two *S. epidermidis* strains (*Figure 2—figure supplement 1*): RP62a, the native CRISPR-Cas10 containing strain, and LM1680 (*Jiang et al., 2013*), a CRISPR-less derivative of RP62a that is more receptive to transformation and conducive to protein purification. To determine the impact of this mutation on crRNA maturation, the entire CRISPR-Cas system was introduced back into LM1680/Δ*pnp* on plasmid p*crispr-cas* (*Hatoum-Aslan et al., 2013*), which encodes a 6-His tag on the N-terminus of Csm2. We then purified Cas10-Csm complexes from the mutant and wild-type strains using Ni$^{2+}$ affinity chromatography, and extracted crRNAs bound to the complexes. While complex formation remained unaffected in the Δ*pnp* mutant (*Figure 2A*), we observed that the crRNA sizes associated with these complexes differed dramatically from the wild-type (*Figure 2B*)—when purified from wild-type background, 6.2 (±2.4) % of complex-associated crRNAs exists in the intermediate state, whereas in the Δ*pnp* mutant, intermediates represent 50.6 (±3.0) % of all crRNAs in the complex (*Figure 2C* and *Figure 2—source data 1*), indicating that PNPase is indeed required for efficient crRNA maturation. To rule out the possibility of an unintentional second-site mutation causing this phenotype, we created a complementation strain in which *pnp* was re-introduced into the genome of the knockout strain, along with several silent mutations in the coding region (denoted as *pnp**) to differentiate between the original wild-type and knock-in strains (*Figure 2—figure supplement 2*). As expected, the crRNA size distribution returned to the wild-type phenotype in the complementation strain (*Figure 2B and C* and *Figure 2—source data 1*), providing confirmation that efficient crRNA maturation relies upon the activity of PNPase.

The persistence of residual maturation in the Δ*pnp* mutant suggests there are probably additional nuclease(s) contributing to this process. In order to identify other maturation nuclease candidates, we consulted the short list of cellular nucleases that were previously found to co-purify with the Cas10-Csm complex (*Walker et al., 2017*). Now, after having established a *bona fide* function for PNPase in crRNA maturation, two other co-purifying nucleases emerged as promising candidates: Ribonucleases J1 and J2. Since PNPase, RNase J1, and RNase J2 have all been found to work together as components of the degradosome in gram positive organisms (*Cho, 2017*; *Redder, 2018*;

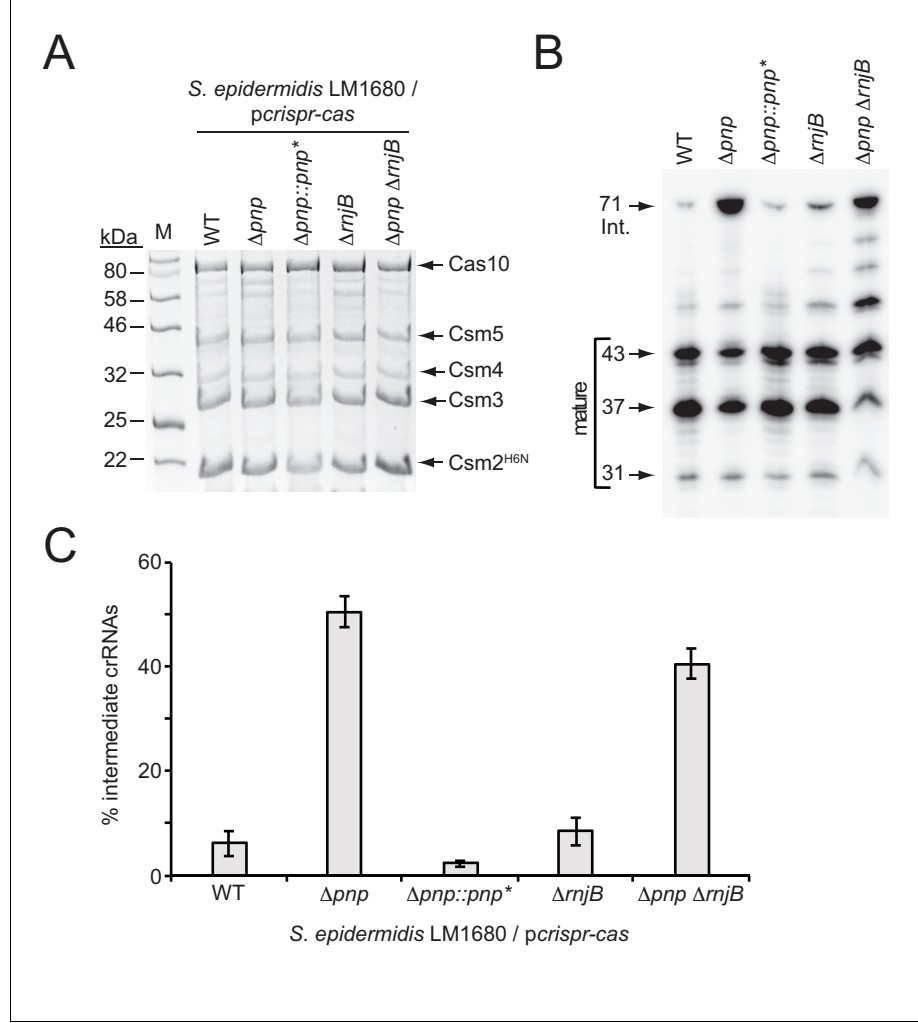

**Figure 2.** PNPase promotes crRNA maturation. (**A**) Cas10-Csm complexes extracted from various strains of *S. epidermidis* LM1680/p*crispr-cas* are shown. The plasmid p*crispr-cas* encodes the entire CRISPR-Cas10 system with a 6-Histidine tag on the N-terminus of Csm2 (Csm2$^{H6N}$). Whole cell lysates from indicated strains bearing this plasmid were subjected to Ni$^{2+}$ affinity chromatography, and purified complexes were resolved and visualized using SDS-PAGE followed by Coomassie G-250 staining. Δ*pnp::pnp\** is a complementation strain in which the Δ*pnp* strain received a variant of the *pnp* gene containing five silent mutations (*pnp\**, see *Figure 2—figure supplement 2* for details). (**B**) Total crRNAs associated with Cas10-Csm complexes purified from indicated *S. epidermidis* LM1680 strains are shown. crRNAs were extracted from the complexes using trizol reagent, radiolabeled on their 5'-ends, and resolved on a denaturing urea-PAGE gel. The gel was exposed to a storage phosphor screen and screens were imaged using a Typhoon phosphor imager. (**C**) Fractions of complex-associated intermediate crRNAs relative to total crRNA content are shown for each strain. Complex-associated crRNAs were analyzed with ImageQuant software to determine band densities. The percent of intermediate crRNAs was calculated as the ratio of the 71 nt band density to the sum of band densities of the major crRNA species (71, 43, 37, and 31 nt). The data represent an average of 3–5 independent trials (±S.D). Specific numbers of replicates are as follows: WT and Δ*pnp*, five replicates; Δ*rnjB*, four replicates; Δ*pnp::pnp\** and Δ*pnp*Δ*rnjB*, three replicates each (see *Figure 2—source data 1*).
DOI: https://doi.org/10.7554/eLife.45393.004

The following source data and figure supplements are available for figure 2:

**Source data 1.** Accompanies *Figure 2C*.
DOI: https://doi.org/10.7554/eLife.45393.007
**Figure supplement 1.** Confirmation of *pnp* and *rnjB* knockouts.
DOI: https://doi.org/10.7554/eLife.45393.005
**Figure supplement 2.** Confirmation of *pnp* knock-in strain.
*Figure 2 continued on next page*

*Figure 2 continued*

DOI: https://doi.org/10.7554/eLife.45393.006

*Roux et al., 2011*), the possibility exists that they might also be working together in the CRISPR-Cas10 immunity pathway. Additionally, the gene encoding RNase J2, *rnjB*, is located directly downstream of *pnp* in the *S. epidermidis* genome, further hinting at a functional link. Ribonucleases J1 and J2 are paralogous enzymes that have been shown to form a complex in the cell and catalyze endonucleolytic and 5′−3′ exonucleolytic cleavage of RNA substrates (*Hausmann et al., 2017*; *Linder et al., 2014*; *Mathy et al., 2007*; *Mathy et al., 2010*; *Raj et al., 2018*). Although the dominant active site in the J1/J2 complex resides in RNase J1, RNase J2 has been shown to cause a synergistic stimulation of RNase J1's nuclease activity through an allosteric mechanism. In order to determine the contribution of these ribonucleases toward CRISPR-Cas10 function, we sought to delete the genes that encode both enzymes. However, despite multiple attempts, we were unable to delete *rnjA*, the gene encoding RNase J1, suggesting this enzyme might be essential for *S. epidermidis* survival. Alternatively, our method for mutagenesis (*Bae and Schneewind, 2006*), which requires cell growth at a range of temperatures, might be incompatible with the extreme temperature sensitivity that has been observed in a *S. aureus* Δ*rnjA* mutant (*Hausmann et al., 2017*). Nonetheless, we were able to create an in-frame deletion of *rnjB*, the gene that encodes RNase J2, in both *S. epidermidis* RP62a and LM1680 strains (*Figure 2—figure supplement 1*). To test for defects in crRNA maturation in the LM1680/Δ*rnjB* mutant, we overexpressed and purified Cas10-Csm complexes from this strain and extracted and visualized the complex-associated crRNAs (*Figure 2*). We observed that Cas10-Csm remains intact when expressed in the Δ*rnjB* background (*Figure 2A*), and the crRNA size distribution is similar to that observed in the wild-type control (*Figure 2B and C* and *Figure 2—source data 1*), indicating that RNase J2 is unlikely to play a significant role in crRNA maturation. To confirm, we also created the Δ*pnp*Δ*rnjB* double-mutant (*Figure 2—figure supplement 1*) and found that it exhibits a maturation defect similar to that observed in the Δ*pnp* single mutant (*Figure 2B and C*). We noted that longer RNAs running between 43 and 71 nucleotides appear more prominently in the double mutant when compared to either single mutant—these species likely represent crRNAs that have undergone incomplete maturation. The more prominent appearance of these bands in the double mutant when compared to the Δ*rnjB* single mutant may imply that RNase J2 plays a modest role in crRNA maturation, and its loss can be compensated by the presence of PNPase. However, since this partial defect in the double mutant does not lead to additional accumulation of the 71 nucleotide intermediate species, we hesitate to speculate further on the significance of these longer crRNAs. From these observations, it can be concluded that PNPase, but not RNase J2, is essential for efficient crRNA maturation, and that there exist other maturation nuclease(s) that have yet to be identified.

## RNaseJ2 is required for CRISPR-Cas10 mediated anti-plasmid and anti-phage immunity

We next wondered whether the activities of PNPase and/or RNase J2 are essential for CRISPR-Cas10 function. To test this, three independent interference assays against diverse mobile genetic elements were conducted (*Figure 3*).

The first is a plasmid challenge assay that measures the transfer efficiency of the conjugative plasmid pG0400 from a *S. aureus* donor into *S. epidermidis* LM1680 recipients bearing p*crispr-cas* (*Figure 3A*). In this system, *spc1* (the first spacer) targets the *nickase* gene in pG0400, therefore, a functional CRISPR-Cas10 system in the recipient strain is expected to lower the conjugation efficiency. Indeed, the controls behaved consistently with previous results (*Hatoum-Aslan et al., 2013*), where the wild-type LM1680 strain bearing the empty vector (pC194) showed a conjugation efficiency in the $10^{-3}$ range, while this same strain harboring p*crispr-cas* exhibited an efficiency in the $10^{-7}$ range (*Figure 3B* and *Figure 3—source data 1*). Interestingly, despite the maturation defect in Δ*pnp*/p*crispr-cas*, this strain had a conjugation efficiency similar to that of the wild-type strain expressing the CRISPR-Cas10 system, suggesting that PNPase is dispensable for anti-plasmid immunity. In contrast, Δ*rnjB*/p*crispr-cas* exhibited a significant defect in immunity, with a conjugation efficiency similar to that of the negative control. Consistent with this result, Δ*pnp*Δ*rnjB*/p*crispr-cas* also

**Figure 3.** RNase J2 is essential for CRISPR-Cas10 immunity against plasmid and phage. (**A**) Illustration of the conjugation assay used to test CRISPR-mediated anti-plasmid immunity. In this assay, the conjugative plasmid pG0400 is transferred from a *S. aureus* RN4220 donor (not shown) into *S. epidermidis* LM1680 recipient strains containing p*crispr-cas*, in which the first spacer (green square) bears complementarity to pG0400. (**B**) The indicated recipient strains harboring either pC194 (empty vector), p*crispr-cas*, or p*crispr-cas-rnjB* (which encodes *rnjB* downstream of the CRISPR-Cas10 system) were mated with *S. aureus* RN4220/pG0400. Cells were subsequently plated on selective media to enumerate *S. epidermidis* recipients and transconjugants. Conjugation (conj.) efficiencies were determined as the average numbers of transconjugants/recipients. Shown is a representative of 2–9 independent trials (see *Figure 3—source data 1* for exact replicate numbers). The significant differences between conjugation efficiencies observed for the wild-type strain in comparison to the Δ*rnjB* and Δ*pnp*Δ*rnjB* mutants are indicated (t-test, one-tailed, p=0.04 and 0.02, respectively). (**C**) Illustration of the phage challenge assay used to test CRISPR-mediated anti-phage immunity in *S. epidermidis* LM1680. In this assay, dilutions of phage CNPx are plated atop lawns of cells containing p*crispr-cas*, in which the second spacer (light blue square) bears complementarity to the phage genome. (**D**) The various strains containing indicated plasmids were challenged with CNPx and the resulting plaque forming units per milliliter (pfu/ml) were enumerated. (**E**) Illustration of the assay to test CRISPR-mediated anti-phage immunity in *S. epidermidis* RP62a. In this assay, the genome-encoded CRISPR-Cas10 system works together with a spacer encoded in the plasmid p*crispr-spcA2* (orange square) to protect against phage Andhra. (**F**) The indicated strains were challenged with Andhra and the resulting pfu/ml were enumerated. For both phage challenge assays, the data shown is an average (±S.D.) of three technical replicates as a representative of several independent trials (see source data files). Dotted lines indicate the limits of detection for these assays.

*Figure 3 continued on next page*

*Figure 3 continued*

DOI: https://doi.org/10.7554/eLife.45393.008

The following source data is available for figure 3:

**Source data 1.** Accompanies *Figure 3B*.
DOI: https://doi.org/10.7554/eLife.45393.009
**Source data 2.** Accompanies *Figure 3D*.
DOI: https://doi.org/10.7554/eLife.45393.010
**Source data 3.** Accompanies *Figure 3F*.
DOI: https://doi.org/10.7554/eLife.45393.011

showed a significant defect in immunity. Interestingly, CRISPR-Cas10 appears to retain a modest level of anti-plasmid immunity in the double mutant, resulting in a conjugation efficiency that is an order of magnitude lower than that observed in the absence of p*crispr-cas* (compare $2.2 \times 10^{-4}$ to $2.6 \times 10^{-3}$). Importantly, complementation strains containing a copy of *rnjB* on p*crispr-cas* (a plasmid called p*crispr-cas-rnjB*) exhibited conjugation efficiencies similar to those of the wild-type strain bearing p*crispr-cas* (*Figure 3B* and *Figure 3—source data 1*). These observations suggest that RNase J2 is essential for CRISPR-Cas10 anti-plasmid immunity.

To determine if this defect applies specifically to plasmid transfer, or if there is a more general phenotype, we tested CRISPR immunity against two unrelated phages. The first, CNPx, is a variant of CNPH82, a temperate phage belonging to the morphological family *Siphoviridae* (*Daniel et al., 2007*; *Maniv et al., 2016*). CNPx can infect *S. epidermidis* LM1680, and *spc2* in p*crispr-cas* bears complementarity to its *cn20* gene (*Figure 3C*). When dilutions of this phage are plated with the LM1680/pC194 negative control, many plaques, or zones of clearing in the bacterial lawn can be observed, approximately $10^7$ plaque-forming units per milliliter (pfu/ml) (*Figure 3D* and *Figure 3—source data 2*). However, when the LM1680/p*crispr-cas* strain is challenged with CNPx, not a single plaque is visible, in agreement with the robust anti-phage immunity previously reported for this system (*Bari et al., 2017*; *Pyenson et al., 2017*). Importantly, when we challenged the mutant strains with CNPx, we observed a significant defect, even more pronounced than that seen for anti-plasmid immunity: While the Δ*pnp* strain appeared to perform like wild-type, the Δ*rnjB* and Δ*pnp*Δ*rnjB* mutations abolished CRISPR-Cas10 function altogether. In addition, returning *rnjB* to these defective mutants on p*crispr-cas-rnjB* restored full immunity against CNPx.

Finally, we wanted to determine if the phenotypes observed thus far are specific to LM1680, or if they can also be seen in the wild-type RP62a strain. *S. epidermidis* LM1680 was originally derived from RP62a as a CRISPR 'escape' mutant which harbors a ~ 258,000 nt deletion encompassing the CRISPR locus and flanking regions (*Jiang et al., 2013*). We wondered if the immunity defect observed in LM1680 might be conditional upon the loss of the regions adjacent to the CRISPR-Cas system. To rule out this possibility, the RP62a Δ*pnp*, Δ*rnjB*, and Δ*pnp*Δ*rnjB* mutants were challenged with phage Andhra (*Cater et al., 2017*) in the presence of the plasmid p*crispr-spcA2* (*Bari et al., 2017*) (*Figure 3E*). Andhra is a lytic phage belonging to the morphological family *Podoviridae*, and *spcA2* targets its DNA polymerase (*gp09*) gene. It is important to note that this system relies upon the RP62a genome-encoded *cas* and *csm* genes, with only the Andhra-targeting spacer being overexpressed on the plasmid. As expected, without p*crispr-spcA2*, all strains are equally susceptible to Andhra, with plaque counts ranging between $10^8$–$10^9$ pfu/ml (*Figure 3F* and *Figure 3—source data 3*). However, when transformed with p*crispr-spcA2*, only the wild-type and Δ*pnp* strains gain complete immunity to Andhra, while the Δ*rnjB* and Δ*pnp*Δ*rnjB* strains remain as sensitive to Andhra as the corresponding negative controls without the p*crispr-spcA2* plasmid. From these collective observations, we can conclude that while PNPase appears dispensable, RNase J2 is essential for CRISPR immunity against diverse mobile genetic elements originating from plasmid and phage. In addition, RNase J2 most likely acts in the CRISPR pathway at a step downstream of crRNA maturation.

## PNPase and RNase J2 promote the efficient clearance of phage-derived nucleic acids

Although CRISPR immunity appeared unperturbed in the Δ*pnp* mutant, we wondered if a partial defect might be present which cannot be resolved by the functional assays. For example, a partial

defect in anti-plasmid immunity might lead to a decrease in plasmid copy number instead of complete elimination, but the copy number reduction might be sufficient to prevent growth on the media that selects for transconjugants. Similarly, a partial defect in anti-phage immunity might result in the survival of a small number of phages, but too few to create a visible plaque in the lawn of bacteria. In order to detect such partial defects in immunity, we used qPCR and qRT-PCR (quantitative PCR and reverse-transcriptase PCR, respectively) to measure DNA and RNA accumulation during an active phage infection. Of the three systems for testing CRISPR function, the RP62a-Andhra system (*Figure 3E*) was selected for this assay for two reasons: First, RP62a is the original clinical isolate containing a single genomic copy of the CRISPR-Cas10 system, thus providing the closest approximation to what might naturally occur. Second, Andhra is strictly lytic and therefore expected to express all genes in a single infection cycle (*Cater et al., 2017*). The latter feature eliminates any gene-specific variations in CRISPR-Cas10 function that would otherwise occur using a temperate phage (*Goldberg et al., 2014*; *Jiang et al., 2016*). In the assay (*Figure 4A*), early-log cells were challenged with Andhra in a 2:1 (bacteria:phage) ratio, and phages were allowed to adsorb to cells for 10 min. Infected cells were then washed with fresh media to remove any phages in suspension. A fraction of the cells was harvested immediately after adsorption (time = 0 min) and every ten minutes thereafter for up to 30 min. Importantly, the latent period for Andhra was previously determined to be 35 min (*Cater et al., 2017*), therefore the selected time points all occur within the timeframe of a single infection cycle before the first burst. Immediately after each harvest, cells were heat-killed to prevent any further changes in nucleic acid content, and then subjected to nucleic acid extraction. The RNA and DNA extracts were analyzed by qRT-PCR and qPCR, respectively, using two pairs of primers: Andhra-specific primers that flank the protospacer region within gene *gp09* (*Figure 4—figure supplement 1*), and genome-specific primers that bind the *gap* gene (encodes glyceraldehyde-3-phosphate dehydrogenase) for normalization. For quantification, the relative abundance of the phage-derived PCR product detected at zero minutes post-infection in RP62a cells was set to one as the reference point.

As expected, in wild-type RP62a cells lacking *spcA2*, phage RNA and DNA levels increased significantly in a time-dependent manner (*Figure 4B and C* and *Figure 4—source datas 1* and *2*); however, the same strain bearing p*crispr-spcA2* exhibited a ten-fold depletion in phage RNA levels, and no increase in phage DNA content by 30 min post-infection. Interestingly, the Δ*pnp*/p*crispr-spcA2* strain showed a significant (~10 fold) accumulation of phage RNA and DNA, but not as high as that observed in the negative control strain devoid of p*crispr-spcA2*. These results suggest that PNPase is indeed required to promote efficient clearance of phage-derived nucleic acids. We also tested the Δ*rnjB*/p*crispr-spcA2* and Δ*pnp*Δ*rnjB*/p*crispr-spcA2* strains, and in agreement with their impaired CRISPR-Cas10 function, both strains showed significant accumulation of phage-derived DNA and transcripts when compared to the wild-type strain bearing p*crispr-spcA2*. Notably, the double mutant showed a greater defect in the clearance of phage nucleic acids than the Δ*rnjB* single mutant, thus confirming the contribution of PNPase towards phage nucleic acid degradation.

One additional observation in this data was striking: The relative amounts of phage nucleic acids in the Δ*rnjB*/p*crispr-spcA2* and Δ*pnp*Δ*rnjB*/p*crispr-spcA2* strains significantly exceeded those seen in the wild-type strain lacking p*crispr-spcA2*. This data implies that PNPase and/or RNase J2 may be acting against phage nucleic acids even in the absence of CRISPR targeting. To test this possibility, we conducted the same time course infection assay in the Δ*pnp* and Δ*rnjB* mutants lacking p*crispr-spcA2* and tracked DNA accumulation (*Figure 4—figure supplement 1* and *Figure 4—figure supplement 1—source data 1*). As expected, the RP62a wild-type control showed that by 30 min post-infection, phage DNA levels rose to ~35 times greater than that seen immediately following adsorption. Strikingly, Δ*pnp* and Δ*rnjB* mutants accumulated over 400 and 1300 times more phage DNA, respectively, by this same time point. These data indicate that PNPase and RNase J2 facilitate the restriction of phage DNA amplification within the cell, even in the absence of a targeting CRISPR system.

## RNase J2 stimulates both RNase and DNase activities of RNase J1

The contributions of PNPase and RNase J2 toward the elimination of phage-derived nucleic acids can occur through multiple mechanisms. One possibility is that these enzymes indirectly prevent phage DNA accumulation by acting strictly as ribonucleases—their swift degradation of phage transcripts might deter the expression of the phage-encoded DNA polymerase, and thus prevent phage

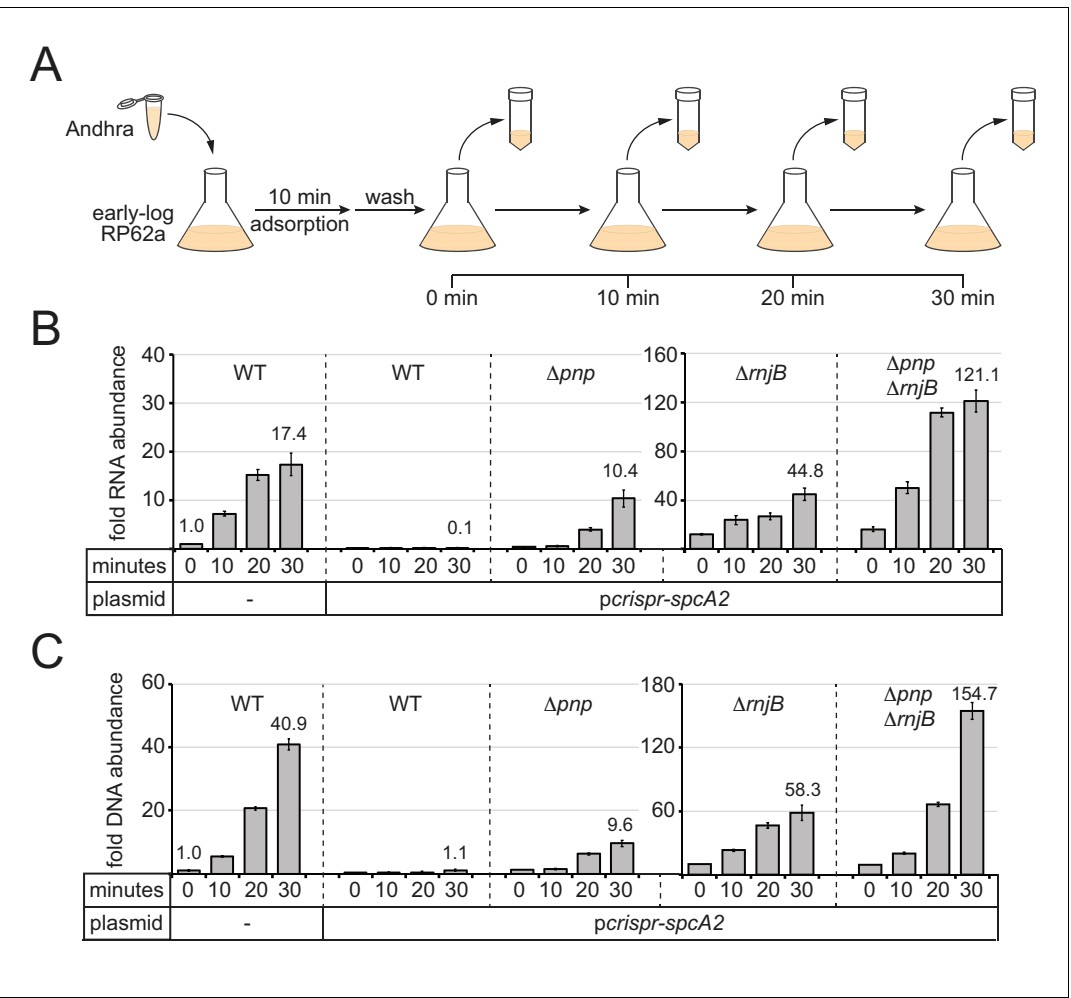

**Figure 4.** PNPase and RNase J2 are required for efficient CRISPR-mediated clearance of invading nucleic acids. (**A**) Diagram of the phage infection assay used to track the accumulation of phage-derived nucleic acids over time. In this assay, early-log *S. epidermidis* RP62a cells were challenged with phage Andhra in a 2:1 (bacteria:phage) ratio. Following a 10 min adsorption period, cells were washed to remove unadsorbed phages. A fraction of infected cells was harvested immediately after the wash (time = 0) and every 10 min thereafter for 30 min. Harvested cells were heat-killed and subjected to RNA or DNA extraction. (**B**) The abundance of phage RNA was quantified for indicated strains at each time point using quantitative reverse-transcriptase PCR (qRT-PCR). Total RNA extracts were reverse-transcribed using primers for the phage-encoded DNA polymerase gene (*gp09*) as well as the host-encoded gene for glyceraldehyde-3-phosphate dehydrogenase (*gap*). A fraction of the resulting cDNA mix was used as a template in the qPCR reaction. Phage cDNA copy number was normalized against host cDNA values, and the normalized value for the 0 min time point in wild-type cells lacking p*crispr-spcA2* was set to one to determine the relative abundance for the rest of the time points. (**C**) The abundance of phage DNA was quantified for indicated strains at each time point using quantitative PCR (qPCR). Total DNA extracts were subjected to qPCR using primers complementary to the phage *gp09* gene and host *gap* gene. Phage DNA copy number was normalized against host values, and the normalized value for the 0 min time point in wild-type cells lacking p*crispr-spcA2* was set to one to obtain the relative DNA abundance for the rest of the time points. For both qRT-PCR and qPCR, the mean ±S.D. of triplicate measurements are shown as a representative of at least two independent trials (see source data files).

DOI: https://doi.org/10.7554/eLife.45393.012

The following source data and figure supplements are available for figure 4:

**Source data 1.** Accompanies *Figure 4B*.
DOI: https://doi.org/10.7554/eLife.45393.015
**Source data 2.** Accompanies *Figure 4C*.
DOI: https://doi.org/10.7554/eLife.45393.016

*Figure 4 continued on next page*

*Figure 4 continued*

**Figure supplement 1.** Accumulation of phage DNA in RP62a strains in the absence of CRISPR-Cas10 immunity.
DOI: https://doi.org/10.7554/eLife.45393.013

**Figure supplement 1—source data 1.** Accompanies *Figure 4—figure supplement 1*.
DOI: https://doi.org/10.7554/eLife.45393.014

DNA replication. Another noncompeting possibility is that phage DNA is degraded directly by one or both of these enzymes. Although PNPase is most well-known for its ribonuclease activity (*Cameron et al., 2018*; *Hui et al., 2014*), we and others have recently shown that bacterial PNPase homologs can also degrade DNA substrates (*Cardenas et al., 2009*; *Walker et al., 2017*). Therefore, it is plausible that the efficient CRISPR-mediated clearance of phage nucleic acids is assisted by PNPase's dual-cleavage of phage DNA and RNA. In contrast, studies on bacterial RNase J homologs have strictly reported ribonuclease activities in these enzymes (*Hausmann et al., 2017*; *Linder et al., 2014*; *Mathy et al., 2007*; *Mathy et al., 2010*). One exception seems to be in two archaeal RNase J homologs, which were shown to catalyze the cleavage of single-stranded DNA (*Levy et al., 2011*). We therefore wondered whether one or both RNase J homologs in *S. epidermidis* might act similarly against DNA substrates.

To test the nucleolytic activities of the *S. epidermidis* RNases J homologs, we overexpressed and purified recombinant RNase J1 and RNase J2 from *E. coli* (*Figure 5A*). In addition, since previous reports have indicated that the dominant active site in an RNase J1/J2 complex resides within RNase J1 (*Mathy et al., 2007*; *Mathy et al., 2010*), we also purified a catalytically-dead variant of RNase J1 (denoted as dJ1) bearing H74A and H76A mutations. We first tested for metal-dependent RNase activity in these enzymes by combining them with a 31 nt RNA substrate (*Figure 5B*) in the presence of various metals (*Figure 5—figure supplement 1*). Consistent with previous reports (*Hausmann et al., 2017*; *Mathy et al., 2010*), we observed insignificant nuclease activity in RNase J2, and $Mn^{2+}$-dependent ribonuclease activity in RNase J1 that leaves behind a single small cleavage product, implying 5′−3′ exonuclease activity. Furthermore, this function appeared more pronounced when RNase J1 was combined with RNase J2 in a 1:1 ratio. To examine the kinetics in more detail, we tracked substrate degradation over time (*Figure 5C and D*), and confirmed that when combined, RNases J1 and J2 act synergistically—whereas RNase J1 alone could only degrade 23 (±14) % of the substrate during a 20 min incubation period, the combined proteins eliminated 95 (±1) % of the substrate in the same amount of time (*Figure 5D* and *Figure 5—source data 1*). Furthermore, the catalytically-dead RNase J1 mutant combined with RNase J2 showed no nucleolytic activity, confirming that the nuclease active site in the J1/J2 complex originates from RNase J1.

We next tested for metal-dependent DNase activity in these enzymes by combining them with a 60 nt DNA substrate (*Figure 5B*) in the presence of various metals (*Figure 5—figure supplement 1*). While we detected little/no nuclease activity from each enzyme alone, a striking $Mn^{2+}$-dependent DNase activity was observed when the enzymes were combined, which again resulted in a single small cleavage product indicative of a 5′−3′ exonuclease function. Additional experiments measuring DNA degradation over time revealed a synergistic activation of DNA cleavage even more pronounced than that observed for RNA cleavage (*Figure 5E and F* and *Figure 5—source data 2*). For example, while RNase J1 alone cleaved 17 (±8) % of the DNA substrate after 5 min, the J1/J2 complex had already degraded 99 (±1) % of the substrate in the same amount of time. In addition to 5′−3′ exonuclease activity, RNase J homologs have also been shown to catalyze endoribonuclease cleavage (*Hausmann et al., 2017*; *Linder et al., 2014*; *Mathy et al., 2007*; *Mathy et al., 2010*). Therefore, we also tested for endonuclease activity against DNA by combining the enzymes alone and in combination with circular DNA substrates, both single-stranded and double-stranded (*Figure 5B*). While cleavage of double-stranded circular or linear DNA could not be detected (*Figure 5G*), we observed degradation of the single-stranded circular DNA by RNase J1, an activity that is significantly stimulated when combined with RNase J2 (*Figure 5H*). Again, the catalytically-dead RNase J1 completely lost this function, indicating that the DNA endonuclease active site resides within RNase J1. Altogether, these observations demonstrate that RNase J2 stimulates both RNase and DNase activities of RNase J1, and therefore lend support to the possibility that these enzymes assist with CRISPR-Cas10 immunity by directly degrading phage DNA and RNA.

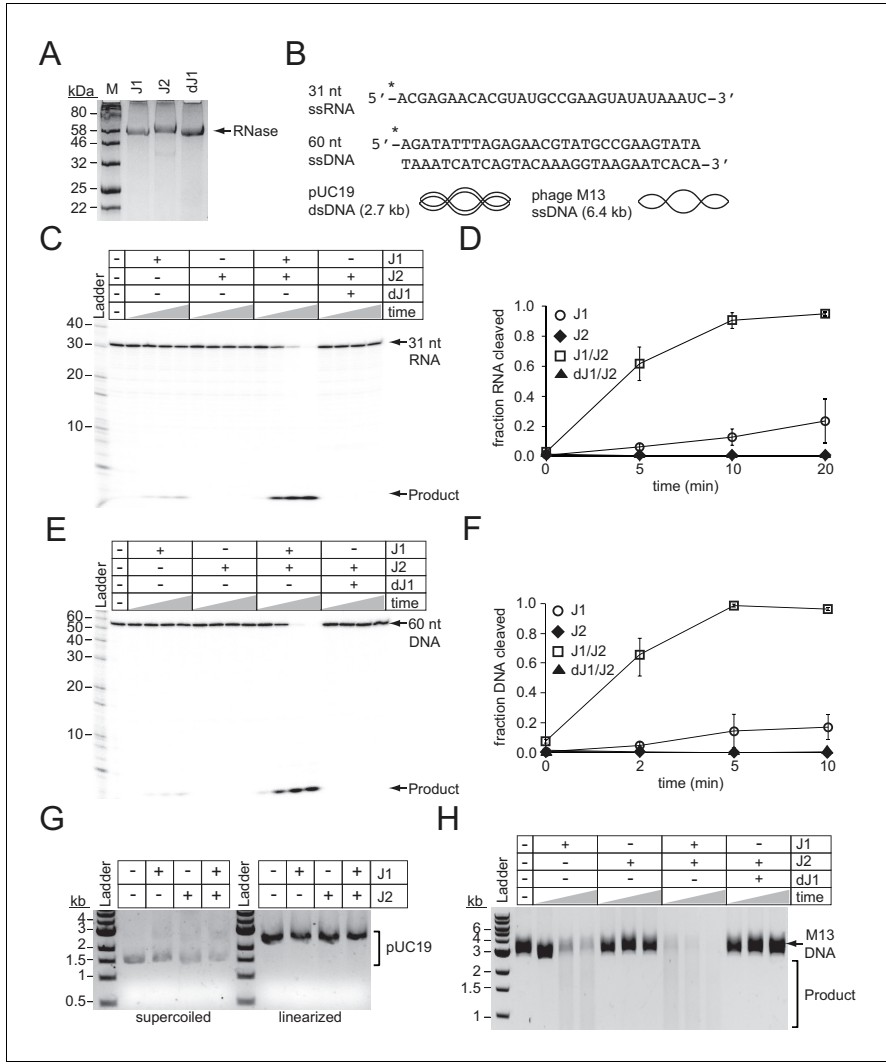

**Figure 5.** RNase J2 promotes RNase and DNase activities of RNase J1. (**A**) Purified recombinant ribonucleases J1, J2, and a catalytically-dead variant of J1 bearing H74A and H76A mutations (dJ1). Proteins were resolved by SDS-PAGE and visualized using Coomassie G-250 staining. (**B**) RNA and DNA substrates used in nuclease assays. (**C**) Ribonuclease activities of RNases J1 and J2. The 5′-end labelled 31-nucleotide RNA substrate was combined with 0.5 pmols of indicated enzyme(s). The reaction mixture was incubated at 37°C for increasing time points (0, 5, 10, or 20 min.) RNAs were resolved using denaturing urea-PAGE. (**D**) Quantification of ribonuclease activity. Shown is the average fraction RNA cleaved (±S.D.) of three independent trials (see **Figure 5—source data 1**). (**E**) Deoxyribonuclease activities of RNases J1 and J2 on linear single-stranded DNA. The 5′-end labelled 60-nucleotide DNA substrate was combined with 0.5 pmols of indicated enzyme(s). The reaction mixture was incubated at 37°C for increasing time points (0, 2, 5, or 10 min.) DNAs were resolved using denaturing urea-PAGE. (**F**) Quantification of deoxyribonuclease activity on single-stranded DNA. The average fraction DNA cleaved (±S.D.) from three independent trials is shown (see **Figure 5—source data 2**). (**G**) Deoxyribonuclease activities of RNases J1 and J2 on double-stranded DNA substrates. The indicated enzymes (two pmols) were combined with 0.1 μg pUC19 DNA that was supercoiled or linearized by PstI digestion. The reaction mixture was incubated at 37°C for 1 hr and DNA was resolved on an agarose gel. (**H**) Deoxyribonuclease activities of RNases J1 and J2 on single-stranded circular DNA. The indicated enzyme(s) (four pmols) were combined with 0.5 μg of phage M13MP18 DNA and the reaction mixture was incubated at 37°C for increasing time points (15, 30, 60 min.) The DNA was resolved on a 0.7% agarose gel. Images in panels G and H are representatives of four independent trials.
DOI: https://doi.org/10.7554/eLife.45393.017

The following source data and figure supplement are available for figure 5:

**Source data 1.** Accompanies **Figure 5D**.
DOI: https://doi.org/10.7554/eLife.45393.019

*Figure 5 continued on next page*

*Figure 5 continued*

**Source data 2.** Accompanies *Figure 5F*.
DOI: https://doi.org/10.7554/eLife.45393.020
**Figure supplement 1.** Metal-dependent cleavage of RNA and DNA by RNases J1 and J2.
DOI: https://doi.org/10.7554/eLife.45393.018

## Discussion

Type III CRISPR-Cas systems rely upon multiple Cas proteins which interact with each other, and also have the potential to recruit and regulate other enzymes with relevant functions to ensure a successful defense. Here, we show that the degradosome-associated nuclease PNPase promotes crRNA maturation (*Figure 2*) and is required for the efficient CRISPR-mediated clearance of invading DNA and RNA (*Figure 4*). We also show that a second degradosome nuclease, RNase J2, is essential for CRISPR immunity against diverse mobile genetic elements originating from plasmid and phage (*Figure 3*). Although bacterial RNase J homologs have been strictly reported as ribonucleases, we present, to our knowledge, the first demonstration of DNA degradation by these enzymes (*Figure 5*). Altogether, our results support a model for CRISPR-Cas10 immunity in which two of the three steps of the pathway, crRNA biogenesis and interference, rely upon the activities of these degradosome-associated nucleases. The possibility remains that PNPase and/or RNase J2 might play indirect roles in CRISPR-Cas10 function, wherein deletion of these degradosome subunits may impact the expression of other host factors that are required for CRISPR function. However, the direct physical interactions that we and others have observed between degradosome components with each other and with the CRISPR machinery (discussed in details below) help to support direct roles for these nucleases in CRISPR immunity (*Figure 6*).

We previously showed that Csm5 binds directly to PNPase and stimulates its nucleolytic activity in a purified system (*Walker et al., 2017*), thus providing mechanisms for PNPase's recruitment and regulation during crRNA maturation and interference in vivo. Since PNPase has been shown to bind RNase J1 (*Raj et al., 2018*; *Roux et al., 2011*), and since RNases J1 and J2 exist as a complex in the cell (*Mathy et al., 2010*), one possible mechanism for the recruitment of RNase J2 to the site of CRISPR interference could be through its connection with PNPase. However, if such an interaction were essential for RNase J2's participation in immunity, then a *pnp* knockout should phenocopy the *rnjB* mutant—this was not the case. Therefore, there may be an additional, more direct interaction between RNase J2 and one or more of the CRISPR-associated proteins that has yet to be identified. Another compatible alternative is that RNase J2 and/or PNPase might locate and degrade phage-derived nucleic acids through a CRISPR-independent mechanism. In support of this notion, we observed that in the absence of a targeting CRISPR-Cas system, the deletion of *pnp* and *rnjB* enables significant phage DNA accumulation which exceeds phage DNA levels in the wild-type strain by 10–40 fold, respectively (*Figure 4—figure supplement 1*). The mechanism by which these degradosome-associated nucleases repress phage DNA replication in the absence of CRISPR interference remains unclear. Interestingly, there have been two reports of phage-encoded proteins that specifically inhibit RNase E, a central subunit of the degradosome in gram negative organisms (*Van den Bossche et al., 2016*; *Hui et al., 2014*; *Marchand et al., 2001*). These observations support the possible existence of a more direct antagonistic relationship between phages and the degradosome that undoubtedly deserves further investigation.

DNA endonuclease activity is particularly important for CRISPR function. In order for phage DNA to get degraded, there must be an initiating endonucleolytic cleavage event that exposes free ends, which are subsequently accessed and processively degraded by exonucleases. Thus far, of the CRISPR-associated nucleases characterized in type III systems, only one has been shown to degrade DNA: Cas10 (*Kazlauskiene et al., 2016*; *Liu et al., 2017*; *Samai et al., 2015*). Furthermore, *S. epidermidis* Cas10 (SeCas10) cleaves only the coding DNA strand in the protospacer region, leaving the non-coding strand unharmed (*Samai et al., 2015*; *Wang et al., 2019*). Although Cas10 homologs in related type III-A systems exhibit nonspecific DNA endonuclease activity (*Kazlauskiene et al., 2016*; *Liu et al., 2017*), this function in SeCas10 has yet to be demonstrated. Therefore, our discovery that RNase J2 is essential for CRISPR-Cas10 immunity, combined with the fact that this enzyme is capable of promoting robust DNA endonuclease cleavage (*Figure 5H*),

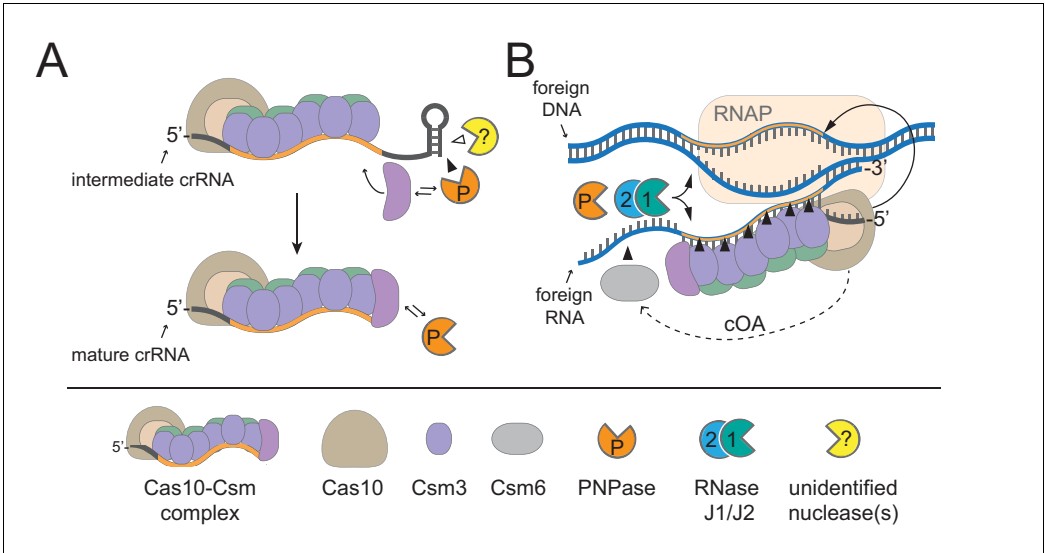

**Figure 6.** A model for CRISPR-Cas10 immunity in which multiple steps in the pathway rely upon degradosome nucleases. (**A**) During complex assembly, Csm5 recruits and stimulates PNPase, which trims the 3'-ends of intermediate crRNAs. The combined activities of PNPase and other unidentified nuclease(s) are required for crRNA maturation. (**B**) During interference, PNPase and the RNase J1/J2 complex help to degrade invading nucleic acids alongside the Cas nucleases. RNAP, RNA Polymerase; cOA, cyclic oligoadenylates.
DOI: https://doi.org/10.7554/eLife.45393.021

opens the possibility that the RNase J homologs assist with CRISPR-Cas10 interference by catalyzing the initiating endonucleolytic cleavage on the non-coding DNA strand. It is worthwhile mentioning that while PNPase exhibits 3'−5' exonuclease activity on DNA and RNA substrates (*Cameron et al., 2018*; *Cardenas et al., 2009*; *Walker et al., 2017*), RNase J homologs catalyze endonuclease activity, as well as 5'−3' exonucleolytic activity, a function only recently thought to be non-existent in bacteria (*Hausmann et al., 2017*; *Linder et al., 2014*; *Mathy et al., 2007*; *Mathy et al., 2010*). Therefore, the combined capabilities of these enzymes provide the full spectrum of nucleolytic activities that are necessary to shred DNA and RNA down to their base parts.

The reliance on non-Cas nucleases by a type III CRISPR-Cas system described herein adds to the sparse collection of similar observations that have been reported for diverse CRISPR-Cas types. The earliest example of such a collaboration was seen in a type II CRISPR-Cas system in *Streptococcus pyogenes*, in which both the crRNA and the tracrRNA are processed by the host-encoded RNase III (*Deltcheva et al., 2011*). In a more recent example, a type III-Bv (variant) system in *Synechocystis* 6803, which lacks a Cas6 homolog, was found to utilize the host-encoded RNase E to catalyze crRNA processing (*Behler et al., 2018*). In yet a third example, a CRISPR-like element in *Listeria monocytogenes* devoid of its own *cas* genes, was shown to rely upon a type I CRISPR-Cas system and host-encoded PNPase for small RNA processing and interference (*Sesto et al., 2014*). From these seemingly disparate observations, perhaps a common theme is beginning to emerge in which diverse CRISPR-Cas types have independently evolved different mechanisms to tap into the arsenal of nucleolytic enzymes that bacteria utilize for routine functions, and channel their activities towards defense in the event of a phage infection. Indeed, such a strategy would help to minimize the energetic costs associated with maintaining dedicated nucleases, while also maximizing the likelihood of a successful defense.

## Materials and methods

### Key resources table

*Continued on next page*

*Continued*

| Reagent type (species) or resource | Designation | Source or reference | Identifiers | Additional information |
|---|---|---|---|---|
| Reagent type (species) or resource | Designation | Source or reference | Identifiers | Additional information |
| Gene (*Stahylococcus epidermidis*) | *pnp* | N/A | GenBank: CP000029.1_SERP0841 | encodes PNPase |
| Gene (*S. epidermidis*) | *rnjB* | N/A | GenBank: CP000029.1_SERP0842 | encodes RNaseJ2 |
| Gene (*S. epidermidis*) | *rnjA* | N/A | GenBank: CP000029.1_SERP0676 | encodes RNase J1 |
| Strain, strain background (*S. epidermidis*) | RP62a | PMID: 3679536 | GenBank: CP000029.1 | LA Marraffini (Rockefeller University) |
| Strain, strain background (*S. aureus*) | RN4220 | PMID: 21378186 | GenBank: NZ_AFGU00000000 | LA Marraffini (Rockefeller University) |
| Strain, strain background (phage CNPx) | CNPx | PMID: 26755632 | GenBank: NC_031241 | LA Marraffini (Rockefeller University) |
| Strain, strain background (phage Andhra) | Andhra | PMID: 28357414 | GenBank: KY442063 | isolated in-house |
| Genetic reagent (*S. epidermidis*) | LM1680 | PMID: 24086164 | | LA Marraffini (Rockefeller University), derivative of RP62a with large deletion |
| Genetic reagent (*S. epidermidis*) | RP62a Δ*pnp* | This paper | | the central 2016 nucleotides of the *pnp* coding region are deleted, *Figure 2—figure supplement 1* |
| Genetic reagent (*S. epidermidis*) | LM1680 Δ*pnp* | This paper | | the central 2016 nucleotides of the *pnp* coding region are deleted, *Figure 2—figure supplement 1* |
| Genetic reagent (*S. epidermidis*) | LM1680 Δ*pnp*::*pnp** | This paper | | a copy of *pnp* with five silent mutations re-introduced into the *pnp* locus, see *Figure 2—figure supplement 2* |
| Genetic reagent (*S. epidermidis*) | RP62a Δ*rnjB* | This paper | | the central 1647 nucleotides of the *rnjB* coding region are deleted, *Figure 2—figure supplement 1* |
| Genetic reagent (*S. epidermidis*) | LM1680 Δ*rnjB* | This paper | | the central 1647 nucleotides of the *rnjB* coding region are deleted, *Figure 2—figure supplement 1* |
| Genetic reagent (*S. epidermidis*) | RP62a Δ*pnp*Δ*rnjB* | This paper | | contains in-frame deletions of *pnp* and *rnjB* as described in the cells above |

*Continued on next page*

*Continued*

| Reagent type (species) or resource | Designation | Source or reference | Identifiers | Additional information |
|---|---|---|---|---|
| Genetic reagent (*S. epidermidis*) | LM1680 Δ*pnp*Δ*rnjB* | This paper | | contains in-frame deletions of *pnp* and *rnjB* as described in the cells above |
| Recombinant DNA reagent | pKOR1 | PMID: 16051359 | | LA Marraffini (Rockefeller University) |
| Recombinant DNA reagent | pKOR1_Δ*pnp* | This paper | | to create in-frame deletion of *pnp* via allelic replacement, see *Supplementary file 1* |
| Recombinant DNA reagent | pKOR1_*pnp*\* | This paper | | to re-insert a copy of *pnp* containing five silent mutations into the *pnp* locus via allelic replacement, see *Supplementary file 1* |
| Recombinant DNA reagent | pKOR1_Δ*rnjB* | This paper | | to create in-frame deletion of *rnjB* via allelic replacement, *Supplementary file 1* |
| Recombinant DNA reagent | pKOR1_Δ*pnp*Δ*rnjB* | This paper | | to create in-frame deletions of *pnp* and *rnjB* via allelic replacement, see *Supplementary file 1* |
| Recombinant DNA reagent | p*crispr-cas* | PMID: 23935102 | | LA Marraffini (Rockefeller University) |
| Recombinant DNA reagent | p*crispr-cas-rnjB* | This paper | | contains the coding region of *rnjB* inserted downstream of *cas6*, see *Supplementary file 1* |
| Recombinant DNA reagent | p*crispr-spcA2* | PMID: 28885820 | | created in-house |
| Recombinant DNA reagent | pET28b-His$_{10}$Smt3 | PMID: 15263846 | | LA Marraffini (Rockefeller University) |
| Recombinant DNA reagent | pET28b-His$_{10}$Smt3-*rnjA* | This paper | | for RNase J1 overexpression and purification, see *Supplementary file 1* |
| Recombinant DNA reagent | pET28b-His$_{10}$Smt3-*rnjB* | This paper | | for RNase J2 overexpression and purification, see *Supplementary file 1* |

*Continued*

| Reagent type (species) or resource | Designation | Source or reference | Identifiers | Additional information |
|---|---|---|---|---|
| Recombinant DNA reagent | pET28b-His$_{10}$Smt3-d*rnjA* | This paper | | for RNase J1/H74A H76A overexpression and purification, see *Supplementary file 1* |
| Recombinant DNA reagent | pUC19 | New England Biolabs | Cat. # N3041S | |
| Recombinant DNA reagent | M13MP18 | New England Biolabs | Cat. # N4040S | |
| Sequence-based reagent | DNA oligonucleotides (multiple) | Eurofins MWG Operon | | To build and sequence recombinant DNA constructs, and to perform quantitative PCR, see *Supplementary file 1* |
| Peptide, recombinant protein | T4 Polynucleotide kinase | New England Biolabs | Cat. # M0201L | |
| Peptide, recombinant protein | Lysostaphin | AmbiProducts via Fisher | Cat. # NC0318863 | |
| Peptide, recombinant protein | DNase I | New England Biolabs | Cat. # M0303S | |
| Peptide, recombinant protein | M-MuLV reverse transcriptase | New England Biolabs | Cat. # M0253L | |
| Peptide, recombinant protein | DpnI | New England Biolabs | Cat. # R0176S | |
| Peptide, recombinant protein | SUMO Protease | MCLAB, http://www.mclab.com/SUMO-Protease.html | Cat. # SP-100 | |
| Commercial assay or kit | EZNA Cycle Pure Kit | Omega Bio-tek via VWR | Cat. # 101318–892 | |
| Commercial assay or kit | EZNA Plasmid DNA Mini Kit | Omega Bio-tek via VWR | Cat. # 101318–898 | |
| Commercial assay or kit | Wizard Genomic DNA Purification Kit | Promega Corporation via VWR | Cat. # A1120 | |
| Commercial assay or kit | PerfeCTa SYBR Green SuperMix | Quanta Biosciences via VWR | Cat. # 101414–150 | |
| Chemical compound, drug | TRIzol Reagent | Thermo Fisher | Cat. # 15596026 | |
| Chemical compound, drug | TRI REAGENT RT | Fisher Scientific | Cat. # NC0850610 | |
| Chemical compound, drug | BAN (4-bromoanisole) | Fisher Scientific | Cat. # NC9734505 | |
| Chemical compound, drug | g-32P-ATP | Perkin Elmer | Cat. # BLU502H250UC | |
| Chemical compound, drug | HisPur Ni-NTA Resin | Thermo Fisher | Cat. # 88222 | |
| Software, algorithm | ImageQuant TL | GE Healthcare/Life Sciences | RRID: SCR_014246 | Version 8.1, used for densitometry |

*Continued on next page*

*Continued*

| Reagent type (species) or resource | Designation | Source or reference | Identifiers | Additional information |
|---|---|---|---|---|
| Software, algorithm | CFX Manager | BioRad | Cat. # 1845000 | Version 3.1, used for qPCR analysis |

## Bacterial strains, phages, and growth conditions

*S. aureus* RN4220 was grown in Tryptic Soy Broth (TSB) medium (BD Diagnostics, NJ, USA). *S. epidermidis* RP62a and LM1680 were grown in Brain Heart Infusion (BHI) medium (BD Diagnostics, NJ, USA). *E. coli* DH5α was grown in Luria Bertani (LB) broth (VWR, PA, USA), and *E. coli* BL21 (DE3) was grown in Terrific broth (VWR, PA, USA) for protein purification. Growth media was supplemented with the following: 10 µg/ml chloramphenicol (to select for p*crispr*-based and pKOR1-based plasmids), 15 µg/ml neomycin (to select for *S. epidermidis* cells), 5 µg/ml mupirocin (to select for pG0400), 30 µg/ml chloramphenicol (to select for *E. coli* BL21 (DE3)) and 50 µg/ml kanamycin (to select for pET28b-His$_{10}$Smt3-based plasmids). Phages CNPx and Andhra were grown on *S. epidermidis* LM1680 and *S. epidermidis* RP62a host cells, respectively. To propagate phages, overnight cultures of each host were diluted 1:100 in BHI supplemented with 5 mM CaCl$_2$ and phage ($1 \times 10^7$ pfu). Cultures were incubated at 37°C with agitation for 5 hr. One-fifth the volume of fresh mid-log host cells was then added into the bacteria:phage culture, and the culture was incubated for an additional 2 hr at 37°C with agitation. Cells were pelleted at 5000 *x g* for 5 min at 4°C, and the supernatant containing phage was passed through a 0.45 µM filter. Phage titer was determined by spotting ten-fold dilutions of the supernatant atop a semisolid layer of 0.5 X heart infusion agar (HIA) medium (Hardy Diagnostics, CA, USA) containing 5 mM CaCl$_2$ and a 1:100 dilution of overnight host culture. Spots were allowed to air dry, and plates were incubated overnight at 37°C. On the following day, plaques were enumerated and phage titers in plaque-forming units/ml (pfu/ml) were determined.

## Construction of pKOR1-based plasmids and transformation into *S. epidermidis* LM1680

The pKOR1 system (*Bae and Schneewind, 2006*) was used to create in-frame deletions of *pnp* and/or *rnjB*, and to re-insert a *pnp* variant (*pnp\**, which has five silent mutations, *Figure 2—figure supplement 2*) into *S. epidermidis* LM1680 and/or RP62a strains. pKOR1_Δ*pnp*, pKOR1_Δ*rnjB*, and pKOR1_Δ*pnp*Δ*rnjB*, which were used to delete indicated genes, were created using a 3-piece Gibson assembly strategy (*Gibson et al., 2009*). Briefly, for pKOR1_Δ*pnp* and pKOR1_Δ*rnjB*, 1.0 kb DNA fragments flanking the gene(s) of interest (upstream and downstream) were obtained via PCR amplification using the *S. epidermidis* RP62a WT genome as template and primers F179, F180, and F331-334 (for pKOR1_Δ*pnp*) and F452-F457 (for pKOR1_Δ*rnjB*) (*Supplementary file 1*). For pKOR1_Δ*pnp*Δ*rnjB*, 1.0 kb DNA fragments flanking both genes were obtained via PCR amplification using the *S. epidermidis* RP62a Δ*pnp* strain as template and primers F452, F459, F455-F457, and F460. For all constructs, the pKOR1 plasmid was used as a template to amplify the backbone with indicated primers. The three PCR products generated for each construct were purified using the EZNA Cycle Pure Kit (Omega Bio-tek, CA, USA) and Gibson assembled. To create the complementation plasmid pKOR1_*pnp\**, a two-piece Gibson assembly was used, wherein the *pnp* gene was amplified from *S. epidermidis* RP62a and assembled with the pKOR1_Δ*pnp* backbone generated with primers L047-L050. All assembled constructs were transformed via electroporation into *S. aureus* RN4220. Four transformants were selected for each construct and confirmed to harbor the plasmid using PCR and DNA sequencing with primers F016, F181, F182, F335, and L009 (*Supplementary file 1*). Confirmed plasmids were purified using the EZNA Plasmid DNA Mini Kit (Omega Bio-tek, CA, USA) and transformed into *S. epidermidis* LM1680 via electroporation. Confirmed *S. epidermidis* LM1680 transformants were used to transfer the pKOR1-based plasmids into *S. epidermidis* RP62a using phage-mediated transduction.

## Transduction of pKOR1-based plasmids into *S. epidermidis* RP62a

The temperate phage CNPx was used to transfer pKOR1-based plasmids from *S. epidermidis* LM1680 into *S. epidermidis* RP62a. Briefly, overnight cultures of *S. epidermidis* LM1680 strains

harboring the plasmids were used to propagate phage CNPx as described above. Filtered phage lysates were then diluted 1:10 in mid-log *S. epidermidis* RP62a cells. The phage:cell mixture was incubated at 37°C for 20 min, and pelleted at 5000 *x g* for 5 min at 4°C. Cell pellets were resuspended in fresh BHI and plated onto BHI agar containing neomycin and chloramphenicol. Plates were incubated overnight at 30°C. Four colonies were picked and confirmed to harbor the plasmid by subjecting them to PCR and sequencing using the appropriate sequencing primers listed in *Supplementary file 1*.

## Generation of *S. epidermidis* mutants

Allelic replacement was used to generate all mutants (*Bae and Schneewind, 2006*). Briefly, *S. epidermidis* strains bearing pKOR1_Δpnp, pKOR1_ΔrnjB, pKOR1_ΔpnpΔrnjB, or pKOR1_pnp* were grown overnight at 30°C. Overnight cultures were streaked onto BHI agar plates containing neomycin and chloramphenicol, and incubated overnight at 43°C to force plasmid integration into the genome. Several colonies were then inoculated into separate tubes containing fresh BHI broth supplemented with neomycin, and incubated overnight at 30°C with agitation in order to force plasmid excision from the genome. The overnight cultures were diluted 100,000 times with sterile water, and 100 µL of diluted culture was plated onto BHI agar plates containing neomycin and anhydrotetracycline (50 ng/µl) and incubated overnight at 37°C to force plasmid loss. Several colonies of different sizes were replica-plated on BHI agar containing neomycin alone and on plates containing neomycin plus chloramphenicol to select against colonies that have lost the plasmid. Only colonies that lost the plasmid (*i.e.* did not grow on the latter plate) were subjected to PCR amplification and sequencing of the PCR product to confirm the presence of the intended mutation (*Figure 2—figure supplement 1* and *Supplementary file 1*). Confirmed mutants were then purified by streaking and selecting single colonies over three consecutive nights.

## Construction of p*crispr-cas-rnjB*

The complementation plasmid p*crispr-cas-rnjB* was constructed using a two-piece Gibson assembly (*Gibson et al., 2009*). Briefly, *rnjB* was amplified from the genome of *S. epidermidis* RP62a using primers L035 and L038 (*Supplementary file 1*) and the backbone was generated in a PCR reaction with p*crispr-cas* (*Hatoum-Aslan et al., 2013*) as template and primers L036 and L039. PCR products were purified using the EZNA Cycle Pure Kit and Gibson assembled. The assembled construct was transformed via electroporation into *S. aureus* RN4220. Four transformants were selected and confirmed to harbor the plasmid using PCR and DNA sequencing with primers L035 and L038. Confirmed plasmids were purified using the EZNA Plasmid DNA Mini Kit and transformed into *S. epidermidis* LM1680 strains via electroporation.

## CRISPR-Cas10 functional assays

Conjugation was carried out exactly as described in *Walker and Hatoum-Aslan (2017)*. Phage challenge assays were carried out by spotting ten-fold dilutions of phage lysate atop a semisolid layer of 0.5 X HIA containing 5 mM $CaCl_2$ and a 1:100 dilution of overnight host culture (wild-type and mutant variants). Spots were air-dried, and plates were incubated overnight at 37°C. On the following day, pfu/ml values were determined. The conjugation and phage challenge data reported represents mean values (±S.D.) of 3–4 technical replicates as a representative of several independent trials (see *Figure 3—source datas 1–3* for exact *n* values).

## Purification of Cas10-Csm complexes

Cas10-Csm complexes containing a 6-His tag on the N-terminus of Csm2 were purified from *S. epidermidis* LM1680/p*crispr-cas* strains using $Ni^{2+}$ affinity chromatography exactly as described in *Chou-Zheng and Hatoum-Aslan (2017)*. Complexes were resolved on a 15% SDS PAGE and visualized with Coomassie G-250. Gels were imaged using the FluorChemR system (Protein Simple, CA, USA).

## Extraction and visualization of crRNAs

Total crRNAs were extracted from purified Cas10-Csm complexes using the TRIzol Reagent (Invitrogen, NY, USA) according to *Hatoum-Aslan et al. (2014)*. Extracted crRNAs were phosphorylated

with T4 Polynucleotide Kinase (New England Biolabs, MA, USA), radiolabeled with γ-[$^{32}$P]-ATP (Perki-nElmer, MA, USA), and resolved on a 15% Urea PAGE gel. The gel was exposed for 10 min to a storage phosphor screen and visualized using a Typhoon FLA 7000 phosphor imager (GE Healthcare Bio-Sciences, PA, USA). For densitometric analysis, the ImageQuant software was used. Percent of intermediate crRNAs was determined with the following equation: ((intensity of intermediate crRNA signal (71 nt) ÷ sum of signal intensities for the dominant crRNA species (71 nt + 43 nt+37 nt+31 nt))×100%). The reported values represent an average of 3–5 replicates (±S.D.), as indicated in the *Figure 2* legend and *Figure 2—source data 1*.

## Phage infection time-course assay

*S. epidermidis* RP62a cells were grown in TSB supplemented with 5 mM CaCl$_2$ to an OD$_{600}$ of 0.3–0.4 and infected with phage Andhra at a multiplicity of infection (MOI) of 0.5. A control culture was also prepared by omitting Andhra. Control and bacteria:phage mixtures were incubated at 37°C for 10 min (without agitation) to allow phage absorption, and cells were pelleted at 5000 *x g* for 5 min. The supernatant was discarded, pellets were resuspended in 40 ml of fresh TSB, and pelleted again as above. Cells pellets were resuspended in 40 ml of fresh TSB. 10 ml of this culture was harvested immediately (time = 0), and the remaining culture was returned to 37°C with agitation. Additional 10 ml aliquots of culture were harvested in a similar manner at time = 10, 20, and 30 min post-infection. Harvested cultures were immediately heat-killed at 95°C for 10 min, pelleted, and washed with 10 ml of sterile water. Final cell pellets were stored at −80°C for nucleic acid extraction.

## Nucleic acid extraction

To extract DNA, cell pellets were resuspended in 100 μl of sterile water, followed by 10 μg of lysostaphin (AmbiProducts, NY, USA) and 5 mM MgCl$_2$. The cell suspension was digested for 2 hr at 37°C. For DNA purification, The Wizard Genomic DNA Purification Kit (Promega Corporation, WI, USA) was used according to the manufacturer's instructions. DNA was stored at 4°C. For RNA extraction, cell pellets were washed twice with 1 ml of TSM Buffer (50 mM Tris-HCl pH 7.5, 0.5 M Sucrose, 10 mM MgCl$_2$) and pelleted at 16,000 *x g* for 1 min. Cells were resuspended in 500 μL of TSM buffer supplemented with 10 μg of lysostaphin and incubated for 20 min at 37°C. Digested cells were pelleted and resuspended in 750 μL of TRI Reagent RT (Molecular Research Center, OH, USA) and 37.6 μL of 4-bromoanisole (Molecular Research Center, OH, USA), and subsequent steps were completed as indicated in the manufacturer's protocol. To eliminate trace genomic DNA, 10 μg of RNA was incubated with 3 units of DNase I (New England Biolabs, MA, USA) for 30 min at 37°C. RNA was extracted with an equal volume of phenol: chloroform: isoamyl alcohol (25:24:1), vortexed for 15 s, and centrifuged at 10,000 *x g* for 2 min. The resulting aqueous phase was mixed with an equal volume of chloroform, vortexed and centrifuged as earlier. The aqueous phase was then combined with 1/10 vol of 3 M NaOAc, pH 5.2, and 3 volumes of ethanol. Mixtures were incubated at −80°C for 30 min to precipitate the RNA. The RNA precipitate was pelleted and washed with 1 ml of 75% ethanol. RNA pellets were then air-dried for 10 min and resuspended in 10 μL of sterile water to obtain a final concentration of 1 μg/μL of RNA.

## cDNA production for qRT-PCR

cDNA was synthesized in 10 μL reactions containing 6 μg of RNA, annealing buffer (5 mM Tris-HCl pH 8.3, 75 mM KCl, 1 mM EDTA), and 1 μM of reverse primer (A475 and S002). Mixtures were incubated at 95°C for 1 min and 48°C for 45 min. The mixtures were then combined with 4 μL of 5X synthesis Buffer (250 mM Tris-HCl pH 8.3, 375 mM KCl, 22.5 mM MgCl$_2$, 75 mM DTT) and 16 μL of Reverse Transcriptase Synthesis Mix (10 mM each of dATP, dCTP, dGTP, dTTP, and 100 U M-MuLV reverse transcriptase), and incubated at 37°C for 30 min. The resulting mix was used as template for qRT-PCR reactions as described below.

## Quantitative PCR

PCR reactions (25 μL) contained 1X PerfeCTa SYBR Green SuperMix (Quanta Biosciences), 0.4 nM of phage-specific primers (A474/A475) or genome-specific primers (S001/S002) (*Supplementary file 1*), and 500 ng of total DNA (for qPCR) or 1 μL of prepared cDNA (for qRT-PCR) as template. Separate standard reactions containing $10^2$–$10^9$ DNA molecules were also prepared using purified phage

DNA extract or bacterial genomic DNA extract. A CFX Connect Real Time PCR Detection System (Bio-Rad, CA, USA) was used to amplify the DNA templates according to the following program: one cycle, 95°C for 3 min; 40 cycles, 95°C for 10 s and 55°C for 30 s. Melt curves were also generated to confirm homogenous product by exposing samples to a final temperature gradient of 65°C to 95°C. Relative fold difference was determined using the equation from the Real-Time PCR Handbook (ThermoFisher Scientific, MA, USA): Fold Difference = $(E_{target})^{Ct\_target} / (E_{normalizer})^{\Delta Ct\_normalizer}$, where $E = 10^{(-1/slope)}$, Ct_target = Ct_target$_{calibrator}$ – Ct_target$_{samples}$, and $\Delta$Ct_normalizer = Ct_ normalizer$_{calibratior}$ – Ct_ normalizer$_{samples}$. The glyceraldehyde-3-phosphate dehydrogenase (*gap*) gene was used for normalization. The Ct value of *S. epidermidis* RP62a at 0 min post-infection was used to calibrate the remaining Ct values. Specific numbers of replicates are found in appropriate figure legends.

## Construction of pET28b-His$_{10}$Smt3-based plasmids

pET28b-His$_{10}$Smt3-*rnjA* and pET28b-His$_{10}$Smt3-*rnjB* were constructed using a two-piece Gibson assembly. Briefly, inserts were obtained by amplifying *rnjA* and *rnjB* from the genome of *S. epidermidis* RP62a and amplifying the backbone from a pET28b-His$_{10}$Smt3 template using primers L030-L033 (for the *rnjB* construct) and L042-L045 (for the *rnjA* construct) (*Supplementary file 1*). PCR products were purified using the EZNA Cycle Pure Kit and Gibson assembled. pET28b-His$_{10}$Smt3-d*rnjA*, encoding a catalytically-dead variant of RNase J1, was constructed via inverse PCR using primers L055 and L056. PCR products were digested with DpnI (New England Biolabs, MA, USA) and purified using the EZNA Cycle Pure Kit. Purified PCR products were 5'- phosphorylated by incubating with T4 Polynucleotide Kinase at 37°C for 30 min, and then circularized by incubating with T4 DNA Ligase (New England Biolabs, MA, USA) overnight at room temperature. Assembled/ligated constructs were introduced into *E. coli* DH5α by chemical transformation. Four transformants for each construct were selected and confirmed to have the plasmid using PCR and DNA sequencing with primers T7P and T7T. Confirmed plasmids were purified using the EZNA Plasmid DNA Mini Kit and introduced into *E. coli* BL21 (DE3) for protein purification.

## Purification of recombinant RNases J1, J2, and dJ1 from *E. coli*

*E. coli* BL21 (DE3) cells harboring the pET28b-His$_{10}$Smt3-based plasmids were grown and induced exactly as previously described (*Walker et al., 2017*). The downstream purification of recombinant proteins from cell pellets had slight modifications. Briefly, cell pellets were resuspended in 30 ml of Buffer A (50 mM Tris-HCl pH 7.0, 1.25 M NaCl, 200 mM Li$_2$SO$_4$, 10% sucrose, 25 mM Imidazole) supplemented with one complete EDTA-free protease inhibitor tablet (Roche), 0.1 mg/ml lysozyme, and 0.1% Triton X-100. Cells were incubated at 37°C for 1 hr and sonicated. Insoluble material was removed via centrifugation and filtration. Cleared lysates were incubated with 3 ml of Ni$^{2+}$-NTA agarose resin (ThermoFisher Scientific, MA, USA) pre-equilibrated with Buffer A, and incubated for 1 hr with constant rotation. The resin was pelleted and washed with 40 ml of Buffer A, followed by a 5 ml wash of 3 M KCl. The resin was then resuspended in 5 ml of Buffer A and transferred to a 5 ml gravity column (G-Biosciences, MO, USA). The resin was further washed with 20 ml of Buffer A. Proteins were eluted stepwise with three aliquots of 1 ml each of IMAC buffer (50 mM Tris-HCl pH 7.0, 250 mM NaCl, 10% glycerol) containing 50, 100, 200 and 500 mM imidazole. Eluted protein fractions were resolved in a 15% SDS PAGE and visualized with Coomassie G-250. The most concentrated fractions (3 ml total) were pooled and mixed with SUMO Protease (MCLAB, CA, USA) and provided SUMO buffer (salt-free). The mixtures were dialyzed for 3 hr against IMAC buffer containing 25 mM imidazole. The dialysate was mixed with 1 ml of Ni$^{2+}$-NTA agarose resin (pre-equilibrated with dialysis buffer), and incubated for 1 hr with constant rotation to collect the His$_{10}$Smt3 tag. The digested dialysate was transferred into a 5 ml gravity column, and the untagged protein was collected in the flow-through. Additional untagged protein was collected by flowing through the column two 500 μL aliquots each of IMAC buffer containing 50, 75, 100 and 500 mM imidazole. Proteins were resolved and visualized as described above. Protein concentrations were determined using absorbance measurements at 280 nm with a NanoDrop2000 spectrophotometer (ThermoFisher Scientific).

## Nuclease assays

Single stranded RNA or DNA substrates were labeled on their 5'-ends by incubating with T4 polynucleotide kinase and γ-[$^{32}$P]-ATP, and purified over a G25 column (IBI Scientific, IA, USA). Labelled substrates were combined with 0.5 pmol of enzymes alone, and in combination, in nuclease buffer (25 mM Tris-HCl pH 7.5, 2 mM DTT) supplemented with 10 mM $MnCl_2$ (unless indicated otherwise). For metal-dependent nuclease activity assays, nuclease reactions were carried out at 37°C for 10 min (ssDNA), or 20 min (ssRNA) using 10 mM of EDTA, $MgCl_2$, $MnCl_2$, $NiCl_2$ and $ZnCl_2$. For time course assays, nuclease reactions were carried out at 37°C for 0, 2, 5 and 10 min (ssDNA), or 0, 5, 10 and 20 min (ssRNA). Reactions were stopped by adding an equal volume of 95% formamide loading buffer and resolved on a 15% Urea PAGE. Gels were exposed to a storage phosphor screen and visualized using a Typhoon FLA 7000 phosphor imager. ImageQuant software was used for densitometric analysis. The fraction of cleaved substrate was determined using the following equation: intensity of cleaved substrate signal ÷ sum of intensities of cleaved plus uncut substrate signals. Values are representative of at least three independent trials. The double-stranded DNA substrate pUC19 (New England Biolabs, MA, USA) was linearized by digesting with PstI (New England Biolabs, MA, USA) and purified using the EZNA Cycle Pure Kit. Circular or linear pUC19 (0.1 μg) was combined in nuclease buffer with 2 pmol of each nuclease alone and in combination. Circular ssDNA (M13MP18) substrate (0.5 μg, New England Biolabs, MA, USA) was combined with 4 pmol of each cellular nuclease alone and in combination, in nuclease buffer. Reactions were incubated at 37°C for 1 hr (dsDNA) or 15, 30 and 60 min (circular ssDNA). Reactions were stopped by incubating on ice for 5 min and digesting with 1 μg of Proteinase K Solution (VWR, PA, USA) for 15 min at room temperature. DNA was resolved on a 0.7% agarose gel containing ethidium bromide, and visualized under UV light using the FluorChemR system. Refer to the appropriate figure legends for specific replicate values.

## Statistical analyses and replicate definitions

All graphed data represent the mean (±S.D.) of *n* replicates, where *n* is indicated in figure legends and source data files. Average values were analyzed using a one-tailed t-test, and p values below 0.05 were considered statistically significant. No statistical methods were used to predetermine sample size. The following terms are used to describe the types of repetitions where appropriate in figure legends and source data files: technical replicate, independent trial, and biological replicate. Technical replicates refer to multiple measurements taken on a single cell line in a single experiment; independent trials refer to repetitions of the same experiment conducted at different times using the same cell lines; and biological replicates refer to independent trials conducted on different lines of cells in which each line comprises a transformant or mutant that was independently generated.

## Acknowledgements

AH-A would like to acknowledge funding for this project from the National Institutes of Health [5K22AI113106-02] and an NSF CAREER award [No. 1749886].

## Additional information

### Funding

| Funder | Grant reference number | Author |
| --- | --- | --- |
| National Institute of Allergy and Infectious Diseases | Career Development Award, 5K22AI113106-02 | Asma Hatoum-Aslan |
| National Science Foundation | CAREER Award, 1749886 | Asma Hatoum-Aslan |

The funders had no role in study design, data collection and interpretation, or the decision to submit the work for publication.

### Author contributions

Lucy Chou-Zheng, Data curation, Formal analysis, Investigation, Methodology, Writing—original draft, Writing—review and editing; Asma Hatoum-Aslan, Conceptualization, Data curation, Formal

analysis, Supervision, Funding acquisition, Visualization, Methodology, Writing—original draft, Project administration, Writing—review and editing

### Author ORCIDs
Asma Hatoum-Aslan  http://orcid.org/0000-0003-2395-8900

### Decision letter and Author response
Decision letter https://doi.org/10.7554/eLife.45393.025
Author response https://doi.org/10.7554/eLife.45393.026

## Additional files

### Supplementary files
• Supplementary file 1. DNA oligonucleotides used in this study. Already corrected in first full submission. Added after initial submission
DOI: https://doi.org/10.7554/eLife.45393.022

• Transparent reporting form
DOI: https://doi.org/10.7554/eLife.45393.023

### Data availability
Source data files for all graphs have been provided.

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
