## [Decision Letter]

Thank you for submitting your article "A Type III-A CRISPR-Cas system employs degradosome nucleases to ensure robust immunity" for consideration by *eLife*. Your article has been reviewed by two peer reviewers and the evaluation has been overseen by a Reviewing Editor and Gisela Storz as the Senior Editor. The reviewers, who have opted to remain anonymous, have recommended your paper for publication, pending minor revisions.

*Reviewer #1:*

In this well-written manuscript, Lucy Chou-Zheng and Asma Hatoum-Aslan demonstrate that PNPase is required for efficient crRNA maturation in the type III-A system from *S. epidermidis*, and that both PNPase and RNase J2 play important roles in degrading foreign nucleic acids. While the function of Cas proteins in this system has received considerable attention, the importance of ancillary factors that complement CRISPR-Cas immune function have not been investigated with the same rigor. This paper adds considerable mechanistic insight and I anticipate that this work will inspire more careful examination of host factors in other immune systems. I have a few very minor suggestions.

Minor comments:

Introduction, first paragraph: Consider adding Yosef et al., 2012. I think this was the first experimental evidence of Cas protein mediated acquisition.

Subsection “RNaseJ2 is required for CRISPR-Cas10 mediated anti-plasmid and anti-phage immunity”, last paragraph, second sentence: I think this sentence is missing a word (i.e., might conditional).

Figure 3 legend: I did not see and explanation for the dashed line in Figure 3D and F.

*Reviewer #2:*

In this manuscript, the authors explore the synergistic activity of the type III-A CRISPR-Cas system and the degradasome complex, which is required for RNA processing and degradation in bacteria. In prior work, the authors provided evidence that several degradasome nucleases (among others, PNPase, RNase J1 and RNase J2) associate with the Csm-Cas10 complex, which mediates type III-A CRISPR-Cas immunity. Here, the authors demonstrate in *Staphylococcus epidermidis* that degradasome nucleases are important for CRISPR RNA (crRNA) maturation and clearance of phage nucleic acids during CRISPR-Cas immunity.

The authors establish three *S. epidermidis* strains *(∆pnp, ∆rnjB* and *∆pnp∆rnjB*) in which either PNPase, RNase J2 or both are deleted (unfortunately the authors could not obtain a knockout of RNase J1, which may be an essential gene). They first tested the importance of these proteins for crRNA maturation and observed that crRNA intermediates accumulate in the absence of PNPase, while RNase J2 is dispensable for crRNA maturation. Next, the authors tested whether PNPase or RNase J2 are required for CRISPR-Cas-mediated immunity using both plasmid conjugation and phage challenge assays. They demonstrate that RNase J2 is required for efficient Csm-Cas10-mediated interference against plasmid conjugation. In a striking set of data, they also show that RNase J2 is required for Csm-Cas10-mediated immunity against phages, and that both PNPase and RNase J2 are required for efficient clearing of phage nucleic acids. Interestingly, they also show that both nucleases may be involved in clearing phage DNA (or in limiting the propagation of the DNA) even in the absence of the CRISPR-Cas system. To support the role of RNase J2, and its partner nuclease RNase J1, the authors performed in vitro RNA and DNA cleavage assays. They convincingly demonstrate that RNase J1 cleaves single-stranded DNA, and that this activity requires RNase J2. These data provide direct evidence to support a role for RNase J1/J2 in phage DNA clearance.

Overall, this is an exceptional study that demonstrates cooperation between a house-keeping complex and the specialized CRISPR-Cas machinery. Other studies have suggested that non-Cas nucleases, including RecBCD, RNase III and PNPase, may be involved in various steps of the immune system. This study provides further compelling evidence toward this emerging trend. The data are very well presented, the manuscript is easy to follow, and the results are reasonably interpreted. The experiments really present a tour-de-force, with the authors anticipating many of my questions. I have a few remaining questions that could likely be addressed through changes to the text.

Minor comments:

1) In Figure 2B, different crRNA intermediates appear to accumulate in the *∆pnp∆rnjB* strain compared to the *∆pnp* strain (running between the 71 and 43 nt band). The authors do not comment on these products. Was this cleavage pattern observed in all replicates? Do the authors have an idea of what these products might be? Although it may be premature to speculate on the origin of these intermediates, the authors should at least note this difference between the two strains in the text, especially if it was reproducible.

2) There is an apparent discrepancy in the data shown in Figure 3 that the authors do not comment on. Conjugation efficiency in the *∆pnp∆rnjB* strain (2.2 x 10^-4^) did not recover to WT levels in the absence of the pCRISPR-Cas plasmid (2.6 x 10^-3^). This suggests that CRISPR-Cas activity is sufficient to provide a low level of plasmid interference (reduction of conjugation efficiency by an order of magnitude) even in the absence of PNPase and RNase J2. However, in the phage challenge assays, the ∆pnp∆rnjB strains displays similar phage sensitivity to strains lacking the CRISPR-Cas system, suggesting that the immune system is not sufficient to provide any immunity against phages in the absence of PNPase and RNase J2. The authors could comment on this apparent difference. Is this simply a consequence of differences in the two types of assays? Could this be related to the apparent CRISPR-independent activity of the degradasome nucleases against phage DNA shown in Figure 5—figure supplement 1?

3) Although the authors data convincingly demonstrate that RNase J2 (and likely J1) cooperate with the CRISPR-Cas machinery, there remains a possibility that deletion of degradasome proteins affects expression of other host factors that may be involved in CRISPR-mediated immunity. The authors should address this possibility in the text.

---

## [Author Response]

Reviewer #1:Minor comments:Introduction, first paragraph: Consider adding Yosef et al., 2012. I think this was the first experimental evidence of Cas protein mediated acquisition.

Done.

Subsection “RNaseJ2 is required for CRISPR-Cas10 mediated anti-plasmid and anti-phage immunity”, last paragraph, second sentence: I think this sentence is missing a word (i.e., might conditional).

Thank you--this error was present in the initial submission, but had been corrected in the full submission (see subsection “RNaseJ2 is required for CRISPR-Cas10 mediated anti-plasmid and anti-phage immunity”, last paragraph).

Figure 3 legend: I did not see and explanation for the dashed line in Figure 3D and F.

As above, this error was present in the initial submission, but had been corrected in the full submission (see Figure 3 legend). The dashed lines in Figure 3D and F indicate the limits of detection for the phage infection assays.

Reviewer #2:Minor comments:1) In Figure 2B, different crRNA intermediates appear to accumulate in the ∆pnp∆rnjB strain compared to the ∆pnp strain (running between the 71 and 43 nt band). The authors do not comment on these products. Was this cleavage pattern observed in all replicates? Do the authors have an idea of what these products might be? Although it may be premature to speculate on the origin of these intermediates, the authors should at least note this difference between the two strains in the text, especially if it was reproducible.

This is an interesting observation that was indeed reproducible across all three replicates. We believe the bands running between 43 and 71 nucleotides in the *∆pnp∆rnjB* strain may represent longer crRNA species that have undergone partial/incomplete maturation. The prominent appearance of these bands in the double mutant, but not the *∆rnjB* single mutant, may imply that RNase J2 plays a modest role in crRNA maturation which can be compensated by the presence of PNPase. We agree that it is worthwhile to note this observation. Accordingly, the following text has been added in the Results section:

“We noted that longer RNAs running between 43 and 71 nucleotides appear more prominently in the double mutant when compared to either single mutant – these species likely represent crRNAs that have undergone incomplete maturation. […] However, since this partial defect in the double mutant does not lead to additional accumulation of the 71 nucleotide intermediate species, we hesitated to speculate further on the significance of these longer crRNAs.”

2) There is an apparent discrepancy in the data shown in Figure 3 that the authors do not comment on. Conjugation efficiency in the ∆pnp∆rnjB strain (2.2 x 10^-4^) did not recover to WT levels in the absence of the pCRISPR-Cas plasmid (2.6 x 10^-3^). This suggests that CRISPR-Cas activity is sufficient to provide a low level of plasmid interference (reduction of conjugation efficiency by an order of magnitude) even in the absence of PNPase and RNase J2. However, in the phage challenge assays, the ∆pnp∆rnjB strains displays similar phage sensitivity to strains lacking the CRISPR-Cas system, suggesting that the immune system is not sufficient to provide any immunity against phages in the absence of PNPase and RNase J2. The authors could comment on this apparent difference. Is this simply a consequence of differences in the two types of assays? Could this be related to the apparent CRISPR-independent activity of the degradasome nucleases against phage DNA shown in Figure 5—figure supplement 1?

There is indeed a difference in the functional assays, in which CRISPR-Cas10 appears to retain some level of anti-plasmid immunity in the *∆pnp∆rnjB* background (Figure 3B), whereas immunity against both phages is completely eliminated in this background (Figure 3D and F). Although it is tempting to speculate that this difference may be due to the CRISPR-independent activity of the degradosome nucleases against phage (Figure 4—figure supplement 1), we hesitate to compare directly between these functional assays due to their inherent differences. For example, immunity against the plasmid and the two phages relies upon different spacer sequences. Additionally, the plasmid and phages exhibit different modes of entry and mechanisms of replication. These variables might impact the efficiency of immunity in unexpected ways. Nonetheless, we agree that it is important to highlight the observed differences in the functional assays. To incorporate these observations, we have added/modified text in the Results section as follows:

“*∆pnp∆rnjB*/p*crispr-cas* also showed a significant defect in immunity. Interestingly, CRISPR-Cas10 appears to retain a modest level of anti-plasmid immunity in the double mutant, resulting in a conjugation efficiency that is an order of magnitude lower than that observed in the absence of p*crispr-cas* (compare 2.2 x 10^-4^ to 2.6 x 10^-3^). “

“[…] we observed a significant defect, even more pronounced than that seen for anti-plasmid immunity: While the *∆pnp* strain appeared to perform like wild-type, the *∆rnjB* and *∆pnp∆rnjB* mutations abolished CRISPR-Cas10 function altogether.”

3) Although the authors data convincingly demonstrate that RNase J2 (and likely J1) cooperate with the CRISPR-Cas machinery, there remains a possibility that deletion of degradasome proteins affects expression of other host factors that may be involved in CRISPR-mediated immunity. The authors should address this possibility in the text.

We completely agree, and have added the following text in the Discussion:

“The possibility remains that PNPase and/or RNase J2 might play indirect roles in CRISPR-Cas10 function, wherein deletion of these degradosome subunits may impact the expression of other host factors that are required for CRISPR function. However, the direct physical interactions that we and others have observed between degradosome components with each other and with the CRISPR machinery (discussed in details below) help to support direct roles for these nucleases in CRISPR immunity (Figure 6).”